# Heavy metal content and microbial characteristics of soil plant system in Dabaoshan mining area, Guangdong Province

**Jianqiao Qin**[1]*, **Xueding Jiang**[2], **Zhiqiang Yan**[3], **Huarong Zhao**[4], **Peng Zhao**[5], **Yibo Yao**[6], **Xi Chen**[7]

**1** Guangdong Provincial Key Laboratory of Environmental Health and Land Resource, School of Environmental and Chemical Engineering, Zhaoqing University, Zhaoqing, 526061, China, **2** School of Environmental and Chemical Engineering, Foshan University, Foshan, 528000, China, **3** Shunde Polytechnic, Shunde, 528300, China, **4** School of Environmental Science and Engineering, Guilin University of Technology, Guilin, 541004, China, **5** South China Institute of Environmental Sciences, Ministry of Ecology and Environment of the People's Republic of China, Guangzhou, 510655, China, **6** Guangdong Provincial Academy of Environmental Scienc, Guangzhou, 510045, China, **7** School of Environmental Science and Engineering, Sun Yat-Sen University, Guangzhou, Guangdong 510275, China

* qinjianqiaosci@126.com

**Data Availability Statement:** All relevant data are within the manuscript and its Supporting Information files.

## Abstract

The disordered mining of Dabaoshan lead-zinc mineral resources in Shaoguan has brought serious harm to the regional ecological environment. In order to investigate the heavy metal pollution status and microbial characteristics of soil plant system in mining area, The distribution of heavy metals in the soil, the activity of soil microorganisms and the accumulation characteristics of heavy metals in the dominant plant *Miscanthus floridulus* were studied. The results indicated that metal element contents of *Miscanthus floridulus* in sequence were: Zn>Pb>Cu> Cd. This study demonstrated that the elemental content of the *Miscanthus floridulus* plant showed Zn>Pb>Cu>Cd, with Zn being the most significantly correlated with soil elements, followed by Pb. Compared with the control group, the *Miscanthus floridulus*-soil system possessed obviously different soil microbial features: intensiver in microbial basal respiration strength, and higher microbial eco-physiological parameters Cmic/Corg and $qCO_2$, but lower in soil microbial biomass. The results showed the soil enzymatic activities decreased significantly with increase of contamination of heavy metals, especially dehydrogenase and urease activities. With the increase of the content of heavy metals in the mining area soil, the intensity of soil biochemical action in the mining area (Q1, Q2) soil decreased significantly, and the biochemical action showed a significant negative correlation with the content of heavy metals in the soil. Compared with the non mining area (Q8) soil, the intensity of soil ammonification, nitrification, N fixation and cellulose decomposition decreased by 43.2%~71.1%, 70.1%~92.1%, 58.7%~87.8% and 55.3%~79.8% respectively. The decrease of soil microbial activity weakened the circulation rate and energy flow of C and N nutrients in the soil of the mining area.

**Funding:** The authors received no specific funding for this work.

**Competing interests:** The authors have declared that no competing interests exist.

## 1. Introduction

The distribution of non-ferrous metals in south China is intensive, and a lot of soil is destroyed by mining or polluted by heavy metals, which has caused serious damage to the soil ecosystem [1,2]. Soil contamination in mining areas can also be transported by surface runoff and biogeochemistry [3–5], endangering the environmental quality of adjacent areas, while contaminated agricultural products can endanger human health through the food chain [6,7]. Therefore, the reclamation of mine land to restore and reuse degraded land has become one of the hot issues in environmental science and soil science [1]. At present, there are few investigations on soil reclamation in non-ferrous metal mines in China, and few studies have been reported on the relationship between soil microbial activity and soil heavy metal content and plant heavy metal accumulation in mining areas [8]. Due to the long-term existence of heavy metal pollution, the microbiota of soil in mining areas had changed significantly, with different levels of sensitivity to heavy metal pollution and significant changes in the biochemical characteristics of soil microorganisms, while some plants had gradually evolved into heavy metal accumulation ecotypes through long-term evolution and natural selection, which had an important role in the reclamation of mining areas [9,10]. Therefore, actively carrying out research in this area to clarify the response mechanism between heavy metal accumulation of plants and soil microbiological and biochemical characteristics in mining areas was important for understanding the nature and causes of soil ecosystem degradation in mines [11–13], and also provides a scientific basis for the search of heavy metal hyperaccumulating plants, which was important for promoting the restoration and reconstruction of mine soil ecosystems [14].

*Miscanthus floridulus* is a perennial herb plant of the Miscanthus, which is widely distributed in southern China and has strong adaptability. It is an ideal plant for phytoremediation due to its rapid growth, large biomass and well-developed root system, which can accumulate heavy metals and reduce their mobility and availability [12,14]. The Dabaoshan Mine is located at the junction of Qujiang District and Wengyuan County in Shaoguan City, Guangdong Province, with a watershed influence of 113°40′-113°43′ E and 24°30′-24°36′N. The region is located in a warm, humid and rainy subtropical monsoon climate with strong weathering of the surface rocks [15,16]. The Dabaoshan Mine is a large iron-polymetallic sulphide associated deposit with a limonite ore in the upper part of the main ore and a copper-sulphide ore in the lower part, with associated non-ferrous ores such as tungsten, bismuth, molybdenum, gold and silver [17–21].

This paper studies the distribution, microbial activity, soil enzyme activity and plant heavy metal accumulation characteristics of the soil ecosystem in Dabaoshan lead-zinc mining area, a typical mining area of mineral resources in northern Guangdong, providing biological basis for the assessment of soil environmental quality and vegetation restoration in the same metal mining area, and providing theoretical basis for the rational use of soil in this area and vegetation restoration in the mining area.

## 2. Materials and methods

### 2.1 Overview of the study area

Beneficiation tailings and waste rock are mainly discharged in two large tailings impoundments (Tielong tailings impoundment and trough-to-pit tailings impoundment) formed by tailings dams and partially flooded throughout the year [22]. The tailings and waste rock contain large amounts of various metal sulphides such as pyrite and chalcopyrite, which are quickly oxidized and form acidic effluent when in contact with the air as a result of various mining activities, while releasing large amounts of toxic and harmful heavy metal ions [23].

The acidic wastewater carrying toxic heavy metal ions was discharged from the two tailings ponds and flows downstream along the river valley to the confluence at Liangqiao and then into the Heng Shi River, where it was joined by water from Chen Gong Wan (a non-mining polluted area) in the middle and lower reaches and eventually flows downstream into the river [24,25]. The acidic wastewater from the tailings pond, which contains a large amount of toxic heavy metal ions, was discharged directly into the river downstream without treatment, causing a serious deterioration of the ecological environment along the mine and its watershed [26–30].

## 2.2 Spot sampling and sample processing

The study region includes the mine waste land, the tailings pond and the Heng Shi River affected by mine wastewater, and the sewage irrigation area downstream to Shangba village. The vegetation in the sample plot was mainly grassland, and the community was dominant, accounting for more than 70%, with 60% ~ 80% coverage [12,15]. The vegetation was flourishing and showed cluster growth. Accordingly, eight sampling points were set up (Table 1), namely, the abandoned land at the top of the Dabaoshan mine [Q1], the hillside of the mine [Q2], the trough-to-pit tailings reservoir [Q3], the downstream tributary of the Tielong tailings reservoir [Q4] (5 km from the mine extraction site), adjacent to the irrigation canal in Shangba Village [Q5] (12 km from the mine extraction site), near the farmland in Xiaba Village [Q6] (16 km from the mine extraction site), the Suoyikeng [Q7] (non-mining contaminated area), Hewu Village [Q8] (non-mining contaminated area). A 1 m x 1 m sample square was set up in different sections for community survey, including plant species, population density, height and cover of *Miscanthus floridulus*.

In addition, 5~10 *M. floridulus* plants were randomly collected in each quadrat and mixed into one sample. Soil samples were collected at a depth of 0–20 cm along an S-shaped route, placed in sterile paper bags and taken back to the laboratory as soon as possible. Part of the fresh soil was ground through a 2 mm sieve, adjusted to the appropriate moisture content and stored at 0–4˚C for testing soil microbiological indicators; the other part of the soil was air-dried and used to determine basic soil physical and chemical properties and heavy metal content. The basic chemical properties were shown in Table 2.

## 2.3 Determination method

Soil and plant heavy metal content: samples of *M. floridulus* were divided into roots, stems and leaves, washed with deionized water, dried to constant weight at 70˚C and crushed through an 80 mesh sieve. The samples were dry ashed and fixed with hydrochloric acid and then determined by atomic absorption spectrophotometry (AA-6880, Shimadzu, Japan). The total

**Table 1. Growth and distribution characteristics of the *M. floridulus* population in Dabaoshan mining area.**

| Mining area | Geographical location | Distance from the main mining area /km | Plant height /cm | Density /(plant·m⁻²) | Coverage /% |
|---|---|---|---|---|---|
| Q1 | N24˚33′36.6″E113˚43′14.0″ | 1 | 210±25.25 | 36 | 80~90 |
| Q2 | N24˚34′06.0″E113˚43′44.3″ | 2 | 200±22.45 | 31 | 70~80 |
| Q3 | N24˚33′32.1″E113˚43′23.8″ | 3 | 190±18.30 | 25 | 60~70 |
| Q4 | N24˚33′38.1″E113˚44′16.8″ | 5 | 178±15.65 | 21 | 55~65 |
| Q5 | N24˚30′14.2″E113˚44′06.9″ | 12 | 165±13.95 | 16 | 50~60 |
| Q6 | N24˚28′46.8″E113˚47′46.1″ | 16 | 150±11.18 | 13 | 40~50 |
| Q7 | N24˚29′57.1″E113˚49′20.8″ | 25 | 145±13.95 | 11 | 40~45 |
| Q8 | N24˚30′03.1″E113˚49′29.6″ | 30 | 141±11.18 | 11 | 40~50 |

Table 2. Some chemical properties of soil samples tested.

| Soil Sample No. | Organic matter (g·kg$^{-1}$) | Effective phosphorus (mg·kg$^{-1}$) | Nitrogen alkali decomposition (mg·kg$^{-1}$) | pH |
|---|---|---|---|---|
| Q1 | 14.649 | 32.255 | 30.1 | 3.15 |
| Q2 | 17.235 | 12.033 | 24.5 | 3.16 |
| Q3 | 2.725 | 36.743 | 63.7 | 3.48 |
| Q4 | 18.012 | 41.233 | 55.6 | 4.04 |
| Q5 | 15.451 | 36.740 | 60.5 | 4.48 |
| Q6 | 19.112 | 50.212 | 63.1 | 4.71 |
| Q7 | 20.652 | 48.568 | 66.4 | 6.55 |
| Q8 | 36.682 | 79.653 | 71.5 | 6.82 |

amounts of soil Zn, Pb, Cu and Cd were digested with HCl, HF and perchloric acid and determined by ICP-OES (Optima5300DV, Perkin-Elmer, Sheldon, CT, USA); the effective state of the corresponding heavy metals in the soil was extracted with 0.1 mol L$^{-1}$ HCl (liquid: soil = 5:1) and the solution to be measured was determined by ICP-OES.

The basic chemical properties of the soil were determined using the soil agrochemical analysis method [30]: soil pH was measured by a pH meter after mixing water and soil at 2.5:1; organic matter was determined using the potassium dichromate volumetric method; nitrogen alkali decomposition was determined using the alkaline diffusion method; and the effective phosphorus was measured using the molybdenum blue colourimetric method [31] after extracting the soil samples with 0.5 mol L$^{-1}$ sodium bicarbonate.

Soil microbial biomass carbon: using the chloroform fumigation-$K_2SO_4$ extraction method [30,32], the organic carbon in the extract was determined using an automatic total organic carbon analyzer (Shimadzu, TOC-500), and the difference between the measured organic carbon values of the chloroform fumigated and unfumigated soil (Fc), divided by the conversion factor Kc (0.45), was obtained as the soil microbial biomass carbon content (Cmic, mg kg$^{-1}$), i.e.: Cmic = Fc/0.45.

Soil basal respiration: measured by closed incubation alkaline absorption titration [33], 20 g of fresh soil sample was weighed into a 500 ml culture flask and the soil was spread evenly flat on the bottom, and the soil water content was adjusted to 60% of the field holding capacity. A 25 ml flask was placed on top of the soil in the flask, then 10 ml of 1 mol L$^{-1}$ NaOH solution was drawn into it, the flask was sealed with a cap and incubated at a constant temperature of 28˚C for 1 month, and the amount of $CO_2$ released was measured every 24 h. Determination of soil biochemical strength [26]: soil ammonification strength was determined by the soil incubation method, and $NH_4^+$-N content was determined by the semi-micro Kjeldahl distillation method; nitrification strength was determined by the solution incubation method, and $NO_3^-$-N reduction was determined by the colorimetric method; nitrification strength was determined by the soil incubation method, and the increase in total soil Nitrogen was analyzed; soil cellulose decomposition strength was determined by the buried slice method, and the weight loss of cloth was analyzed.

Soil enzymatic activity was determined using the literature [30]: 3,5-dinitrosalicylic acid colorimetric method for soil sucrose enzyme activity [34]; sodium phenol colorimetric method for soil urease activity [35,36]; sodium phosphate colorimetric method for soil acid phosphatase activity [37]; potassium permanganate titration method for soil catalase activity [38–40]; ninhydrin colorimetric method for soil protease activity [41]; and TTC colorimetric method for soil dehydrogenase activity [42].

## 2.4 Data processing

Statistical analysis of data is performed using a combination of Microsoft Excel 2003 and SPSS 16.0 software and the significance of differences between means were analyzed using Duncan's multiple comparisons (SSR test, $p < 0.05$).

# 3. Results and discussion

## 3.1 Heavy metal content of soil-plant systems in the Dabaoshan Pb-Zn mining area

**3.1.1 Analysis of heavy metal content in soils.**   The results of the analysis of the total and effective contents of Zn, Pb, Cu and Cd in the test soils (Table 3) show that the contents of Zn, Pb, Cu and Cd in the soils of six sampling points in the mine area were all above the secondary standard for soil environmental quality (GB15618-1995). The Zn, Pb, Cu and Cd contents of the soil in the mine area gradually decreased from the centre of the mine area to the periphery of the mine area, and the corresponding total mean values were 10. 24, 19. 98, 119.10 and 21.35 times higher than those of the control soil samples (soil samples No. 7 and 8). 47.57 times higher than the control soil samples (soil samples 7 and 8), indicating that the soils in the mining area were all contaminated with heavy metals to different degrees. The data in Table 2 showed significant differences in the total and effective amounts of Zn, Pb, Cu and Cd in each soil sample from the centre of the tailings pond to the periphery by the LSD0.05 test ($P < 0.05$). This difference may be related to the degree of mixing of tailings sand with soil in different zones under the action of external camp forces (mainly in surface runoff water erosion). The closer to the tailings pond, the higher the total tailings deposition and the correspondingly higher heavy metal content in the soil [28,43].

**3.1.2 Heavy metal element content of different parts of the *Miscanthus floridulus* plant.**   As shown in Table 3, the accumulation of Zn, Pb, Cu and Cd in the shoot of *M. floridulus* increased significantly with the increase of soil heavy metal content. See Table 4 for the heavy metal content in different parts of the *M. floridulus*. According to Table 4, The average contents of Zn, Pb, Cu and Cd in *M. floridulus* were as follows: Zn(110.21±18.12 mg·kg$^{-1}$) > Pb(71.11±5.97 mg·kg$^{-1}$) > Cu(50.15±5.37 mg·kg$^{-1}$) > Cd(3.02±0.81 mg·kg$^{-1}$). The content of elements in roots, stems and leaves was Zn > Pb > Cu > Cd. The content of Zn in roots was (123.25±14.10) mg·kg$^{-1}$, which was 1.6 times that of Pb, 2.4 times that of Cu and 35.9 times

**Table 3. Heavy metal contents of soil samples collected in the mine area (mg·kg⁻¹).**

| Soil Sample No. | Heavy metals full amount | | | | Active state heavy metal content | | | |
|---|---|---|---|---|---|---|---|---|
| | Zn | Pb | Cu | Cd | Zn | Pb | Cu | Cd |
| Q1 | 1768.25 | 1241.13 | 1698.75 | 9.13 | 291.75 | 68.5 | 238.5 | 1.42 |
| Q2 | 780.88 | 1000.5 | 1572.5 | 5.11 | 15.07 | 56.43 | 127.2 | 0.09 |
| Q3 | 640.50 | 886.25 | 893.63 | 1.88 | 53.95 | 98.8 | 149.48 | 0.47 |
| Q4 | 440.25 | 750.25 | 695.20 | 1.25 | 50.25 | 90.35 | 105.5 | 0.40 |
| Q5 | 295.25 | 602.75 | 288.75 | 0.63 | 68.83 | 138.03 | 47.79 | 0.51 |
| Q6 | 250.33 | 350.25 | 150.25 | 0.55 | 40.25 | 58.30 | 40.25 | 0.30 |
| Q7 | 161.75 | 83.63 | 20.1 | 0.25 | 3.58 | 7.61 | 5.7 | 0.14 |
| Q8 | 60.50 | 35.13 | 6.38 | 0.13 | 1.56 | 6.03 | 1.33 | 0.01 |
| a* | 250.00 | 200.00 | 50.00 | 0.30 | - | - | - | - |

Note: a*, Soil Environmental Quality Class II Standard (GB15618-1995).

**Table 4. Content of elements in different parts of *M. floridulus* (mg·kg$^{-1}$).**

| Soil sample No. | Zn | | | Pb | | | Cu | | | Cd | | |
|---|---|---|---|---|---|---|---|---|---|---|---|---|
| | root | stem | leaf | root | stem | leaf | root | stem | leaf | root | stem | leaf |
| Q1 | 195.21 | 93.78 | 85.17 | 160.18 | 56.08 | 36.35 | 82.5 | 51.15 | 43.28 | 30.22 | 18.22 | 6.45 |
| Q2 | 143.25 | 72.33 | 52.19 | 121.56 | 37.04 | 13.28 | 71.1 | 27.29 | 21.21 | 13.55 | 11.25 | 3.51 |
| Q3 | 123.15 | 62.75 | 30.15 | 81.16 | 15.05 | 5.25 | 52.1 | 7.25 | 3.25 | 3.52 | 1.22 | 0.41 |
| Q4 | 94.55 | 53.44 | 11.18 | 52.36 | 8.01 | 4.65 | 32.6 | 5.26 | 2.22 | 1.56 | 0.29 | 0.29 |
| Q5 | 85.28 | 42.65 | 8.11 | 21.15 | 4.03 | 3.25 | 22.3 | 3.35 | 1.25 | 0.52 | 0.22 | 0.15 |
| Q6 | 76.29 | 31.85 | 3.16 | 11.33 | 1.05 | 0.75 | 11.0 | 2.55 | 0.23 | 0.33 | 0.20 | 0.11 |
| Q7 | 72.19 | 30.75 | 3.03 | 9.30 | 1.01 | 0.63 | 10.79 | 2.33 | 0.20 | 0.31 | 0.21 | 0.13 |
| Q8 | 71.11 | 29.85 | 2.96 | 8.93 | 1.05 | 0.61 | 10.05 | 2.18 | 0.19 | 0.26 | 0.17 | 0.11 |
| Average content | 110.21±18.12 | | | 71.11±5.97 | | | 50.15±5.37 | | | 3.02±0.81 | | |

that of Cd, respectively. The contents of Zn, Pb, Cu and Cd were different in different parts of the plant, showing root > stem > leaf.

**3.1.3 Relationship between *M. floridulus* and soil heavy metals.** The *M. floridulus* in relation to the soil elements were analyzed as shown in Table 5. The results showed that the correlation between Zn in all parts (root, stem and leaf) and soil Zn was significant (p<0.05), with the correlation between root and stem being highly significant (p<0.01); the correlation between Cu and Cd in root and soil was significant; and the correlation between Pb in stem was highly significant. It indicated that the correlation of Zn in all parts of the *M. floridulus* was better than the correlation of Pb, Cu and Cd elements, and that the uptake and accumulation of Zn by the plant increases with the increase of soil Zn content. It also shows that even the same plant did not absorb and accumulate different elements differently, reflecting the characteristics of the plant itself on the one hand, and the influence of soil elements on the plant and their ability to migrate in the plant on the other [44,45].

## 3.2 Variations in the microbiological characteristics of the Dabaoshan lead-zinc mining area

The changes in soil microbial characteristics parameters were shown in Fig 1. It can be seen that from Q8 and Q7 (control soil samples) to Q1, as the heavy metal content increased, the more luxuriant the growth of *M. floridulus*, the higher the rate of basal microbial respiration, while the soil microbial biomass tended to decrease, and the t-test showed a significant difference between them (p<0.05). Fig 1 also showed that two microbial physiological and ecological parameters, the ratio of soil microbial carbon to total organic carbon (microbial quotient = Cmic/Corg) and the ratio of soil basal respiration intensity to microbial carbon (metabolic quotient $qCO_2$ = Rmic/Cmic), showed similar trends, with significant increases in both parameters from Q8 to Q1 in control soils (p<0.05). This indicated that the metabolic

**Table 5. Correlation coefficient between the content of elements in *M. floridulus* and that in soils.**

| Parts of plant | Zn | Pb | Cu | Cd |
|---|---|---|---|---|
| Root | 0.725** | 0.171 | 0.489* | 0.425* |
| Stem | 0.621** | 0.528** | 0.365 | 0.025 |
| Leaf | 0.465* | 0.321 | 0.320 | 0.116 |

Note: *p≤0.05

**p≤0.01, n = 23.

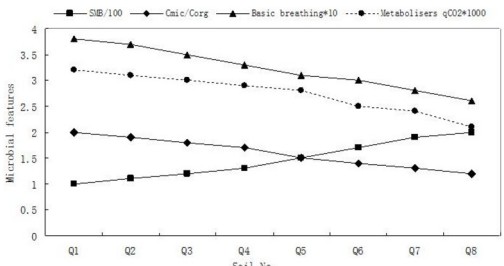

**Fig 1. Variation of microorganisms features from different sites (SMB = Soil Microbial Biomass).**

capacity of microorganisms was significantly altered in the soil-plant system of the Pb-Zn mine. This may be related to the dense growth of *M. floridulus* in the above-ground part.

As seen from the results of the study, the microbial quotient (Cmic/Corg) and metabolic quotient ($qCO_2$) of the Pb-Zn mining soil-*M. floridulus* plant system showed an increasing trend compared to the control soil, and the soil respiration rate was significantly enhanced, which may be caused by a significant increase in the intensity of microbial activity due to the resistance of microorganisms to long-term stress by heavy metals (especially Cu) [46,47]. In contrast, soil microbial biomass showed a decreasing trend, probably due to microorganisms maintaining a low substrate utilization rate [48], indicating that the magnitude of soil microbial biomass in the Pb-Zn mining soil-*M. floridulus* system did not correspond well to the intensity of microbial physiological activity and that the metabolic capacity of microorganisms was significantly altered. In such an environment, microorganisms need to consume more energy in order to maintain their normal life activities, resulting in less efficient use of energy carbon by soil microorganisms [49].

Table 6 showed a comparison of the correlations between microbial parameters. Under normal conditions, microbial respiration intensity was positively correlated with microbial biomass, so the soil respiration rates of the control soil samples (Q7 and Q8) in Table 5 were positively correlated with microbial load, but this correlation became weaker in the mine site (Q6 to Q1) soils, indicating that the soil microbial load did not correspond well to its respiration intensity in the Pb-Zn mine soil- *M. floridulus* system. Similarly, the negative correlations of Cmic/Corg and $qCO_2$ with microbial biomass show that lower microbial loads in Pb-Zn mining soils correspond to higher microbial physiological activity [23,50]. Although Cmic/Corg was usually negatively correlated with soil organic carbon, and the organic carbon content of the mine (Q1-Q6) soils was significantly lower than that of the control soils, the negative correlation between Cmic/Corg and Corg (organic carbon) becomes increasingly poor from the control soils (Q7 and Q8) to the mine (Q1-Q6) soils, reflecting the fact that in a soil ecosystem as specific as the Pb-Zn mine, the Cmic/Corg and Corg (organic carbon) correlations become increasingly poor. It also reflected the fact that the intensity of soil microbial physiological activity had changed significantly in such a special soil ecosystem as the Pb-Zn mine [51].

**Table 6. Correlation between soil microbial characterisation parameters (n = 3).**

| Item | Q1 | Q2 | Q3 | Q4 | Q5 | Q6 | Q7 | Q8 |
|---|---|---|---|---|---|---|---|---|
| Cmic/Corg & SMB | -0.825 | -0.755 | -0.625 | -0.447 | -0.208 | -0.101 | -0.059 | -0.053 |
| $qCO_2$ & SMB | -0.881 | -0.813 | -0.795 | -0.719 | -0.638 | -0.311 | 0.072 | 0.075 |
| Basic Breathing & SMB | 0.475 | 0.619 | 0.664 | 0.773 | 0.820 | 0.882 | 0.901 | 0.911 |
| Cmic/Corg & Corg | -0.568 | -0.643 | -0.755 | -0.785 | -0.814 | -0.828 | -0.939 | -0.950 |

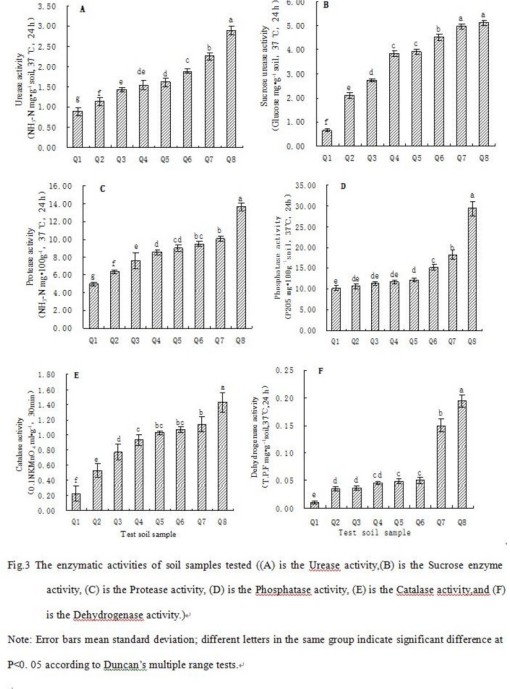

Fig.3 The enzymatic activities of soil samples tested ((A) is the Urease activity,(B) is the Sucrose enzyme activity, (C) is the Protease activity, (D) is the Phosphatase activity, (E) is the Catalase activity,and (F) is the Dehydrogenase activity.)

Note: Error bars mean standard deviation; different letters in the same group indicate significant difference at P<0.05 according to Duncan's multiple range tests.

**Fig 2.** The enzymatic activities of soil samples tested ((A) is the Urease activity, (B) is the Sucrose enzyme activity, (C) is the Protease activity, (D) is the Phosphatase activity, (E) is the Catalase activity,and (F) is the Dehydrogenase activity). Note: Error bars mean standard deviation; different letters in the same group indicate significant difference at P<0.05 according to Duncan's multiple range tests.

### 3.3 Analysis of soil enzyme activity in the Dabaoshan Pb-Zn mining area

The results of the analysis of the six enzyme activities of each test soil were shown in Fig 2. As can be seen from Fig 2, there was a certain degree of variation in enzyme activity between the soil samples, with the enzyme activity of the soil in the tailings contaminated area being lower and the control soil (Q8) having the highest values for all the enzymes, with the urease, protease and dehydrogenase activities showing greater variation, followed by phosphatase activity and less variation in sucrase and peroxidase activities. Fig 2 also showed that there was also a large variation in the magnitude of the different types of enzyme activities in the mine soils. The mean values of urease, dehydrogenase, protease, phosphatase, sucrase and catalase activities were 43.81%, 27.40%, 57.86%, 44.19%, 83.04% and 58.16% of those of the control soil respectively.

The results of the factor analysis of the heavy metal content and basic chemical properties of the test soils were shown in Table 7, which yielded a variance contribution of 98.60% for the first principal factor of heavy metal element content, indicating that this principal factor can

**Table 7. Factor analysis of heavy metal contents and chemical properties of soil.**

| Factor | Zn | Pb | Cu | Cd | Organic matter (g·kg⁻¹) | Effective phosphorus (mg·kg⁻¹) | Nitrogen alkali decomposition (mg·kg⁻¹) | pH |
|---|---|---|---|---|---|---|---|---|
| Contribution of variance % | 69.448 | 16.418 | 11.360 | 1.374 | 1.176 | 0.206 | 0.018 | 0.000 |
| Cumulative contribution of variance% | 69.448 | 85.866 | 97.226 | 98.600 | 99.776 | 99.982 | 100.000 | 100.000 |

reflect the variability of the whole variable system of the test soils, which showed that the main influencing factor leading to the reduced enzyme activity of the soil in this tailings contaminated area was the heavy metal element content. This was related to the inhibitory effect of heavy metals on enzymes [51–53]. The mechanism of action may be due to the combination of the active site of the enzyme molecule, the sulfhydryl group, and the imidazole-containing ligand, forming a more stable complex, resulting in a competitive inhibition with the substrate, or it may be due to the inhibition of the growth and reproduction of soil microorganisms by heavy metals, reducing the synthesis and secretion of enzymes in the organism and finally leading to a decrease in soil enzyme activity [54,55].

### 3.4 Intensity of soil biochemical action

The intensity of soil ammonification, nitrification, Nitrogen fixation and cellulose decomposition was carried out with the direct participation of all major physiological groups of soil microorganisms, and the soil played an important role in maintaining the C and N balance of its ecosystem with the coordination of these microbial communities [56]. The intensity of soil biochemical action was usually considered as one of the comprehensive indicators of soil microbial activity [57]. The results of the analysis (Table 8) showed that the intensity of soil biochemical action in the mining area (Q1 and Q2) soils was significantly reduced, compared with the non-mining area (Q7 and Q8) soils, in which the intensity of soil ammonification, nitrification, Nitrogen fixation and cellulose decomposition decreased by 43.2%~71.1%, 70.1%~92.1%, 58.7%~87.8% and 55.3%~79.8%, which shown a significant negative correlation ($p<0.05$) with the content of heavy metals in the soil, leading to a weakened rate of nutrient cycling of C and N elements in the soil [58], thus reducing the intensity of the supply of effective nutrients in the soil, resulting in the current situation where the above-ground part grows only with a large number of heavy metal-tolerant plants (*M. floridulus*). The mechanism of tolerance deserves further study [59,60].

## 4. Conclusions

1. The content of heavy metal elements in the soil of lead-zinc mining area was higher than that of the control soil, and the soil of the mining area was polluted to different degrees. The heavy metal content in the dominant species of *M. floridulus* in the mining area was significantly positively correlated with the total and available content of heavy metal in the soil. From the mine contaminated area to the control area, the heavy metal content in soil and plant gradually decreased.

**Table 8. Strength of biochemical action of soil at different levels of contamination g kg$^{-1}$.**

| Soil Sample No. | Ammonification | Nitrification | Nitrogen fixation | Cellulosic decomposition strength |
|---|---|---|---|---|
| Q1 | 0.104±0.013c | 0.013±0.006e | 0.025±0.008d | 1.21±0.080f |
| Q2 | 0.125±0.014c | 0.023±0.007e | 0.045±0.031d | 1.55±0.201e |
| Q3 | 0.147±0.020b | 0.041±0.011d | 0.065±0.022c | 2.17±0.103d |
| Q4 | 0.197±0.020b | 0.048±0.018d | 0.085±0.012c | 2.67±0.105d |
| Q5 | 0.215±0.079b | 0.068±0.010c | 0.105±0.015c | 3.16±0.150c |
| Q6 | 0.255±0.010b | 0.099±0.012b | 0.155±0.019b | 4.66±0.200b |
| Q7 | 0.348±0.020a | 0.165±0.011a | 0.206±0.020a | 5.96±0.031a |
| Q8 | 0.358±0.020a | 0.175±0.012a | 0.216±0.020a | 5.98±0.041a |

Note: Different alphabets after the data in the same column indicate significant differences ($p < 0.05$), n = 3.

2. In the soil of lead-zinc mining area, the intensity of soil microbial physiological activity has changed obviously. With the increase of soil heavy metal content, soil microbial biomass carbon decreased gradually, and soil microbial biomass carbon was significantly correlated with the total and available state of polluted heavy metals, while soil basic respiration and microbial metabolism quotient increased with the increase of soil heavy metal content. Compared with non-mining soil, the activities of several soil enzymes in mining soil were inhibited to a certain extent.

3. Decreased soil microbial activity was one of the most important signs of damage to soil eco-systems in mining areas and one of the most important factors in the ecological evolution of soil microbes in mining areas. Reduced soil microbial activity weakens the cycling rate and energy flow of C and N nutrients in mine soils. The results show that in the restoration of a heavy metal polluted soil ecosystem, not only the above-ground vegetation should be restored, but also the soil microbial ecological community should be restored and the soil plant ecosystem should be rebuilt.

## Supporting information

**S1 Data.**
(XLS)

## Author Contributions

**Conceptualization:** Huarong Zhao.

**Data curation:** Zhiqiang Yan.

**Formal analysis:** Xueding Jiang, Zhiqiang Yan, Yibo Yao.

**Investigation:** Xi Chen.

**Methodology:** Peng Zhao.

**Project administration:** Jianqiao Qin.

**Writing – review & editing:** Jianqiao Qin.

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
