## [Decision Letter · Decision Letter 0]

23 Jan 2023

PONE-D-22-31232Microbial characteristics of soil-Miscanthus floridulus plant system in Dabaoshan Lead-zinc mining area, northern Guangdong ProvincePLOS ONE

Dear Dr. QIN,

Thank you for submitting your manuscript to PLOS ONE. After careful consideration, we feel that it has merit but does not fully meet PLOS ONE’s publication criteria as it currently stands. Therefore, we invite you to submit a revised version of the manuscript that addresses the points raised during the review process.

We look forward to receiving your revised manuscript.

Kind regards,

Muhammad Aamer Mehmood, Ph.D.

Academic Editor

PLOS ONE

Journal Requirements:

 “Special Project of Key Areas of Colleges

and Universities in Guangdong Province  (Science and Technology Promoting Rural Revitalization)

“Research and Development of Key Technologies for Resource Utilization of Manure from Large-

Scale Livestock and Poultry Breeding in Rural Areas of Western Guangdong” (No.:2021ZDZX4023)”

5. We note that Figure 1 in your submission contain map images which may be copyrighted. All PLOS content is published under the Creative Commons Attribution License (CC BY 4.0), which means that the manuscript, images, and Supporting Information files will be freely available online, and any third party is permitted to access, download, copy, distribute, and use these materials in any way, even commercially, with proper attribution. For these reasons, we cannot publish previously copyrighted maps or satellite images created using proprietary data, such as Google software (Google Maps, Street View, and Earth). For more information, see our copyright guidelines: http://journals.plos.org/plosone/s/licenses-and-copyright.

6. We note you have included a table to which you do not refer in the text of your manuscript. Please ensure that you refer to Table 1 in your text; if accepted, production will need this reference to link the reader to the Table.

Reviewers' comments:

Reviewer's Responses to Questions

**Comments to the Author**

1. Is the manuscript technically sound, and do the data support the conclusions?

Reviewer #1: Yes

Reviewer #2: Yes

2. Has the statistical analysis been performed appropriately and rigorously? 

Reviewer #1: Yes

Reviewer #2: No

3. Have the authors made all data underlying the findings in their manuscript fully available?

Reviewer #1: Yes

Reviewer #2: Yes

4. Is the manuscript presented in an intelligible fashion and written in standard English?

Reviewer #1: Yes

Reviewer #2: Yes

5. Review Comments to the Author

Reviewer #1: The manuscript entitled “Microbial characteristics of soil-Miscanthus floridulus plant system in Dabaoshan Leadzinc mining area, northern Guangdong Province” is interesting. However, I have added some corrections in the PDF manuscript file and suggest the following points that may help to improve the work:

Please rewrite title in more simple and effective way.

The introduction should be extended to discuss related literature, hypothesis and research aim in details.

Please explain the basis of using Miscanthus floridulus as test plant species.

The discussion section still needs improvement, and should be linked to the findings of the previous reports on this topic.

Conclusion section should be concise with future direction of research work at the end.

Reviewer #2: I reviewed the manuscript titled ‘Microbial characteristics of soil-Miscanthus floridulus plant system in Dabaoshan Lead-zinc mining area, northern Guangdong Province’. The study is interesting and built a set of data to draw logical conclusions of reclamation and phytoremediation of contaminated mining areas; however, I feel that the authors need to add substantial improvements in the ms before the publication is approved. My comments are as follows:

1) Title seems to be incomplete since the microbial characteristics were not the only set of parameters studied by the authors in this study.

2) The abstract lacking the objective of the study. The authors need to state what is the rational of this study in the abstract

3) I don’t know why there’s no line and page numbering in the ms making things difficult for reviewer to spot and convey the mistakes

4) Introduction Line 1: South of the China

5) Introduction is too short and faulty. The authors focused a plant species Miscanthus floridulus in the title and mentioned nothing about this plant in the introduction. Moreover, authors may also introduce the readers about Dabaoshan Lead-zinc mining area in the introduction section

6) The objective and hypothesis of the study at the end of the introduction is all over the place e.g., the authors in the current never made an attempt to collect super accumulated plant germplasm resources. Authors need to focus on what they have experimented only

7) Too long introduction of the study area in M&M section. This may be summarized here and introductory sentences may be shifted to introduction

8) “Miscanthus floridulus community was dominant, accounting for more than 70%, with 60% ~ 80% coverage” This sentence is without any reference. Again authors are putting extra information in M&M some of which may be shifted to Introduction with references

9) In the ms, Miscanthus floridulus is written without italicized at many places. Please take care

10) In M&M some instruments e.g., Atomic Absorption Spectrophotometer is mentioned without make and model

11) Why the authors measured Zn, Pb, Cu, Cd with ICP-OES, why not the other heavy metals too?

12) 3.1.3 M. floridulus on the relationship between mango plants and soil heavy metal elements……I am unable to understand what is mango here?

13) The legends of the figures are embedded in the text without the figures in place; however they are also presented at the bottom with figures in place. I think remove them from the text

14) I feel that the authors presented the results strongly but discussion is partly weak. Please take care

15) Why there is no statistical analysis in table 3 and 4?

16) I am not expert in English language; however, I spotted some typographical and syntax errors in the ms which authors can double check

6. PLOS authors have the option to publish the peer review history of their article (what does this mean?). If published, this will include your full peer review and any attached files.

Reviewer #1: No

Reviewer #2: No

---

## [Author Response · Author response to Decision Letter 0]

19 Apr 2023

Response to Reviewer 1 Comments

Point 1: Please rewrite title in more simple and effective way.

Response 1: Thank you for your valuable comments. We have changed the title of the paper to: “Heavy metal content and microbial characteristics of soil plant system in Dabaoshan mining area, Guangdong Province”.

Point 2: The introduction should be extended to discuss related literature, hypothesis and research aim in details.

Response 2: Thank you for your advice. The relevant literature, hypotheses and research objectives are further discussed in the introduction.

Point 3: Please explain the basis of using Miscanthus floridulus as test plant species.

Response 3: Thank you very much for your constructive comments. Here is an explanation added to the text,

 “Miscanthus floridulus is a perennial herb plant of the Miscanthus, which is widely distributed in southern China and has strong adaptability [14,15]. It is an ideal plant for phytoremediation due to its rapid growth, large biomass and well-developed root system, which can accumulate heavy metals and reduce their mobility and availability [16,17].”

Point 4: The discussion section still needs improvement, and should be linked to the findings of the previous reports on this topic.

Response 4: Thank you very much for your constructive comments. The results of the experiment and the discussion part were done together; This time, the discussion section has been strengthened based on the test results.

Point 5: Conclusion section should be concise with future direction of research work at the end.

Response 5: Thank you very much for your constructive comments. The conclusions have been reviewed and revised to add directions for future work. Please look at the revised paper, thank you.

Response to Reviewer 2 Comments

Point 1: Title seems to be incomplete since the microbial characteristics were not the only set of parameters studied by the authors in this study.

Response 1: Thank you for your comments. The title of the paper has been revised to " Heavy metal content and microbial characteristics of soil plant system in Dabaoshan mining area, Guangdong Province".

Point 2: The abstract lacking the objective of the study. The authors need to state what is the rational of this study in the abstract

Response 2: Thank you for your advice. Here the abstract is reviewed and modified, and the purpose of the study is added.

Point 3: I don’t know why there’s no line and page numbering in the ms making things difficult for reviewer to spot and convey the mistakes

Response 3: Thank you for your advice. Here the paper has been checked and modified to make it more standardized.

Point 4: Introduction Line 1: South of the China

Response 4: Thank you for your suggestion. The problem has been solved. The first sentence has been modified here, please see the modification trace, thank you.

Point 5: Introduction is too short and faulty. The authors focused a plant species Miscanthus floridulus in the title and mentioned nothing about this plant in the introduction. Moreover, authors may also introduce the readers about Dabaoshan Lead-zinc mining area in the introduction section

Response 5: Thank you for your advice. Here, in the introduction part, the introduction of the research on Miscanthus floridulus is added. In addition, the introduction of Dabaoshan lead-zinc mining area is added in the introduction.

Point 6: The objective and hypothesis of the study at the end of the introduction is all over the place e.g., the authors in the current never made an attempt to collect super accumulated plant germplasm resources. Authors need to focus on what they have experimented only

Response 6: Thank you for your advice. Here, the research objectives and assumptions at the end of the introduction are reasonably revised to make them more close to the actual situation of this study. Thank you.

Point 7: Too long introduction of the study area in M&M section. This may be summarized here and introductory sentences may be shifted to introduction

Response 7: Thank you for your comments. Here, the introduction of the study area is simplified.

Point 8: “Miscanthus floridulus community was dominant, accounting for more than 70%, with 60% ~ 80% coverage” This sentence is without any reference. Again authors are putting extra information in M&M some of which may be shifted to Introduction with references

Response 8: Thank you for your comments. References have been added here, please refer to the revised paper for details.

Point 9: In the ms, Miscanthus floridulus is written without italicized at many places. Please take care

Response 9: Thank you for your advice. Here, the corresponding phrase is italicized and checked. Thanks a lot.

Point 10: In M&M some instruments e.g., Atomic Absorption Spectrophotometer is mentioned without make and model 

Response 10: Thank you for your advice. This time, the manufacturer and model of the atomic absorption spectrophotometer are given here.

Point 11: Why the authors measured Zn, Pb, Cu, Cd with ICP-OES, why not the other heavy metals too?

Response 11: Thank you for your advice. Here, according to the previous research results, the most polluted heavy metals in Dabaoshan mining area are "zinc, lead, copper and cadmium", so only these four heavy metals have been measured.

Point 12: 3.1.3 M. floridulus on the relationship between mango plants and soil heavy metal elements……I am unable to understand what is mango here?

Response 12: Thank you for your advice. Here is a clerical error. The sentence has been revised as follows: Relationship between M. floridulus and soil heavy metals

Point 13: The legends of the figures are embedded in the text without the figures in place; however they are also presented at the bottom with figures in place. I think remove them from the text

Response13: Here, the corresponding questions have been modified. 

Point 14: I feel that the authors presented the results strongly but discussion is partly weak. Please take care

Response 14: Thank you for your advice. This is revised. Please see the revised paper for details.

Point 15: Why there is no statistical analysis in table 3 and 4?

Response 15: Thank you for your advice. Because the difference is not obvious and is not the focus of the study, there is no difference test.

Point 16: I am not expert in English language; however, I spotted some typographical and syntax errors in the ms which authors can double check

Response 16: Thank you for your advice. This is revised. Please see the revised paper for details.

---

## [Editor Report · Decision Letter 1]

24 Apr 2023

Heavy metal content and microbial characteristics of soil plant system in Dabaoshan mining area, Guangdong Province

PONE-D-22-31232R1

Dear Dr. Jianqiao QIN,

We’re pleased to inform you that your manuscript has been judged scientifically suitable for publication and will be formally accepted for publication once it meets all outstanding technical requirements.

Kind regards,

Muhammad Aamer Mehmood, Ph.D.

Academic Editor

PLOS ONE

Additional Editor Comments (optional):

Dear Authors

I am happy to see the revisions, and recommending this manuscript for publication in PLOS ONE

However, there are minor language corrections needed, which should be corrected during proofreading carefully

Additionally, please make sure Miscanthus floridulus is italicized throughout the manuscript

All units are as per SI format and in consistency with each other throughout the manuscript

Congratulations

With best wishes

Aamer
---

## [Editor Report · Acceptance letter]

2 Jun 2023

PONE-D-22-31232R1 

Heavy metal content and microbial characteristics of soil plant system in Dabaoshan mining area, Guangdong Province 

Dear Dr. Qin:

I'm pleased to inform you that your manuscript has been deemed suitable for publication in PLOS ONE. Congratulations! Your manuscript is now with our production department. 

Kind regards, 

on behalf of

Dr. Muhammad Aamer Mehmood 

Academic Editor

PLOS ONE